# Copolymers Derived from Two Active Esters: Synthesis, Characterization, Thermal Properties, and Reactivity in Post-Modification

**DOI:** 10.3390/molecules27206827

**Published:** 2022-10-12

**Authors:** Thi Phuong Thu Nguyen, Nadine Barroca-Aubry, Caroline Aymes-Chodur, Diana Dragoe, Gaëlle Pembouong, Philippe Roger

**Affiliations:** 1Institut de Chimie Moléculaire et des Matériaux d’Orsay (ICMMO), UMR 8182, Université Paris-Saclay, CNRS, 91405 Orsay, France; 2Equipe Chimie des Polymères, Institut Parisien de Chimie Moléculaire, Sorbonne Université, CNRS, 4 Place Jussieu, 75005 Paris, France

**Keywords:** copolymerization, RDRP, thermal properties, reactivity, post-modification, methacrylate derivatives

## Abstract

Copolymers with two distinguished reactive repeating units are of great interest, as such copolymers might open the possibility of obtaining selective and/or consequent copolymers with different chemical structures and properties. In the present work, copolymers based on two active esters (pentafluorophenyl methacrylate and *p*-nitrophenyl methacrylate) with varied compositions were synthesized by Cu(0)-mediated reversible deactivation radical polymerization. This polymerization technique allows the preparation of copolymers with high to quantitative conversion of both comonomers, with moderate control over dispersity (*Đ* = 1.3–1.7). Additionally, by in-depth study on the composition of each copolymer by various techniques including elemental analysis, NMR, FT-IR, and XPS, it was possible to confirm the coherence between expected and obtained composition. Thermal analyses by DSC and TGA were implemented to investigate the relation between copolymers’ composition and their thermal properties. Finally, an evaluation of the difference in reactivity of the two monomer moieties was confirmed by post-modification of copolymers with a primary amine and a primary alcohol as the model.

## 1. Introduction

Developments in polymer post-modification have been effective in response to increasing demand for highly functional and complex polymer structures. Substitution of polymers bearing active esters provides one approach to polymer post-modification that satisfies the concept of ideal post-polymerization modification, i.e., fast, efficient, straightforward, and applicable on a large scale [1]. Until now, several activated esters have been exploited, for example *N*-hydroxysuccinimide esters [2,3], pentafluorophenyl-containing (meth)acrylates [4,5,6], nitrophenyl (meth)acrylates [7,8], glycidyl (meth)acrylates [9,10,11,12], and recently *N*-(4-vinylphenyl)sulfonamides [13]. The homopolymers of these activated esters, notably pentafluorophenyl-containing (meth)acrylates, can be modified sequentially with different molecules of interest to prepare copolymers for various applications [14,15,16]. Meanwhile, research has also exploited differences in reactivity of different reactive monomers to synthesize copolymers whose post-modification can be performed sequentially for individual types of monomer [17,18]. Therefore, it is of great interest to obtain other new copolymers derived from repeating units of different chemical reactivity. 

Recently, we reported that Cu(0)-mediated RDRP is a facile, labor-reduced polymerization technique to synthesize the homopolymer of pentafluorophenyl methacrylate (PFPMA) [19]. This relatively new technique uses stable salts and zero-valent metals to avoid the direct use of oxygen/air sensitive salts like Cu(I) halides, hence reducing labor work and simplifying the polymerization setup [20]. Its advantages make Cu(0)-mediated RDRP an adaptable and efficient methodology for the synthesis of polymers, not only in solution but also from supporting matters varying from inorganic [21,22], organic [23,24,25], to biological substrates [26,27]. However, copolymerization using Cu(0)-mediated RDRP has rarely been reported in the literature. Herein, we report the synthesis and detailed structural characterization of copolymers derived from pentafluorophenyl methacrylate (PFPMA) and *p*-nitrophenyl methacrylate (NPMA) by Cu(0)-mediated RDRP (NMR, ATR FTIR, elemental analysis, and XPS). Thermal properties of the obtained copolymers were also investigated by TGA and DSC analysis. Furthermore, copolymers of different compositions were subjected to chemical post-modification to directly evaluate the reactivity of PFPMA and NPMA moieties towards *n*-butyl amine and citronellol—a primary alcohol. 

## 2. Results and Discussion

### 2.1. Synthesis of Copolymers of PFPMA and NPMA

Copolymers of PFPMA and NPMA were synthesized by Cu(0)-mediated RDRP with variation of molar fractions of comonomers, as shown in Figure 1. copoxNyF-zz refers to the difference in reaction condition, in which x/y = f_NPMA_/f_PFPMA_ indicates molar ratio in feed between NPMA and PFPMA, and zz indicates the reaction times of the copolymerization, 19 h or 68 h. 

Even though the conditions for Cu(0)-mediated RDRP of PFPMA have been well established [19], the total contradictory difference in solubility of the two homopolymers (i.e., PPFPMA and PNPMA) leads to the need for changing polymerization conditions to obtain a homogeneous reaction, which has been reported to be crucial for controllably synthesizing PPFPMA by this technique. The copolymerization was carried out using THF:sulfolane mixture (2:1 *v*/*v*), because PNPMA has low solubility in THF, while PPFPMA has poor solubility in sulfolane.

Table 1 summarizes the synthesis of different copolymers with variation in molar fraction of the two comonomers.

In this work, we considered that the comonomers of PFPMA and MMA tend to form random copolymers. Previous studies have shown that copolymerization of the PFPMA-MMA [28] and NPMA-MMA [29] systems lead to reactivity ratio products r_1_ × r_2_ of 0.47 and 0.87, respectively. These values are characteristic of a random sequence. As the substituents of the methacrylate function of PFPMA and of NPMA are substantially identical in sizes, and both are electro-acceptors, it is presumed that the resulting copolymer should also exhibit a random sequence. From results presented in Table 1, molar ratios of PFPMA in obtained copolymers were close to that in feed, indicating a good compatibility during polymerization and close reactivity by Cu(0)-mediated RDRP of the two comonomers.

By increasing the polymerization time from 19 h to 68 h (Table 1), higher conversion was achieved for both comonomers, yet molar ratio in feed and in final copolymers did not show significant change. SEC analyses of these copolymers (Appendix A) showed moderate control over polymerization for all conditions (*Đ* < 1.5) except for copo1N1F-68 (*Đ* = 1.71). Experimental *M*_n SEC_ were systematically greater than the theoretical M_n theo_, with *M*_n SEC_/M_n theo_ ratios varying between 1.6 and 2.3. The two studied ester-substituents of copolymer were larger than those of PMMA, it therefore seems consistent that the copolymers obtained in this work would have larger hydrodynamic volume than PMMA with the same M_n theo_, leading to a higher M_n SEC_. 

Although we established optimal conditions for homopolymerization of PFPMA by Cu(0)-mediated RDRP [19], the introduction of NPMA into the polymerization system appeared problematic. This was confirmed by the copolymerization of PFPMA and NPMA under optimized conditions of Cu(0)-mediated RDRP, with MBPA as initiator and dNbpy as ligand (Table 1). Upon addition of NPMA to the reaction, the dispersity of the obtained copolymer increased tremendously to *Đ* = 2.59 (Table 1), obtained by conventional calibration in THF (+TEA 1 g.L^−1^). It is seen that under similar conditions, acquired PPFPMA showed good control with *Đ* = 1.05 (Table 1). On the other hand, the homopolymerization of NPMA with MBPA and dNbpy showed poorer control in terms of polydispersity (Table 1). The moderate control over dispersity in the obtained copolymers might result from the presence of PNPMA, which appears in the literature to be able to form complexation with CuBr_2,_ hence disrupting the fast reaction between the propagating chains and the deactivator CuBr_2_ [8].

Figure 1 provides the ^1^H-NMR spectra of copolymers at different compositions in THF-d_8_. It can be seen that protons of the backbone appear at 0.8–1.9 ppm (methyl protons, symbolized by the red rectangles) and 1.9–3.0 ppm (methylene protons, blue triangle). Protons from the *p*-nitrophenyl pendant group (orange star) respond at around 7.4 ppm and 8.2 ppm. In addition, signals were recorded coming from the methylene group at the α-extremity (green ball) at 4.1 ppm, indicating the conservation of end-chain generated by an initiator. From these ^1^H-NMR spectra, the ratio between the comonomer units in each copolymer was also calculated and shown good agreement with expected the ratio obtained by calculation from conversions. DOSY measurements of the three copolymers obtained after 68 h of polymerization (Appendix A) indicates that the obtained products were copolymers composed of both NPMA and PFPMA moieties rather than a mixture of two homopolymers. 

### 2.2. FTIR Analysis: Chemical Environment Analysis

ATR-FTIR was used to validate the chemical environment in the acquired copolymers. The spectra of two homopolymers (PPFPMA and PNPMA) and three copolymers are presented in Figure 2a. Bond assignments of major peaks observed in these spectra are listed in Appendix A. The copolymers composed of PFPMA and NPMA possessed characteristic bands for their two major components, including bands corresponding to nitro groups (C-N at 1205 cm^−1^ and N-O at 1350 cm^−1^), bending signals of aromatic C-H (862 cm^−1^) in NPMA, as well as stretching of the C-F bond (993 cm^−1^) in PFPMA. Interestingly, carbonyl vibration bands of PPFPMA and PNPMA were not completely superposed; the C=O band of PFPMA was located at a higher wavenumber in PPFPMA (1778 cm^−1^) than in PNPMA (1752 cm^−1^), presumably due to a higher conjugation effect caused by pentafluorophenyl compared with the *p*-nitrophenyl group. Consequently, C=O and C-O bands of copolymers also shifted compared with those of homopolymers, due to the incorporation of both NPMA and PFPMA, correlated to NPMA and PFPMA composition. Figure 2b presents enlargement sections of peaks corresponding to vibration of C=O and C-O bonds. Figure 2c shows the correlation between peak maxima with molar proportion of PFPMA in copolymers. It can be seen from these two figures that C=O band shifted to a higher wavenumber as the quantity of pentafluorophenyl increased, while the opposite trend was observed for C-O bands.

Furthermore, Figure 2d shows ratios between areas of bands characterized for C-F of PFPMA and C-H of NPMA as functions of theoretical molar percentage of PFPMA. The peak areas of C-N, C-F, and C-H bonds were calculated by integration of corresponding bands by Origin software. Information on the range and baselines used for these calculations is described graphically in Appendix A. As expected, when the amount of PFPMA in the copolymer increased, the ratio of the peak between C-F/C-H bonds increased linearly. Nonetheless, comparison between peak areas of C-F and N-O bonds as in Figure 2e also revealed the same trend with the same linearity. These results confirmed in a qualitative manner that the compositions of the obtained copolymers were close to the expected values. 

### 2.3. X-ray Photon Spectroscopy: Details on Composition of Copolymers

XPS analysis of three copolymers including copo1N2F-68, copo1N1F-68, and copo2N1F-68 was investigated to determine their composition, in comparison with other techniques such as NMR and elemental analysis. The use of XPS is of interest as it allows the quantification of ratios between different atoms and different bonds independently, hence providing in-depth information on the structure of the copolymers. 

The atomic number percentages of elements composing the copolymers were calculated from surveyed spectra, as shown in Figure 3. XPS quantification results are summarized in Appendix A, in comparison with results from NMR and elemental analysis. 

As seen from Appendix A, the atomic number percentages obtained by NMR and XPS were very similar. More importantly, the atomic weight ratios of C/N, C/O, and F/N, as presented in Appendix A, prove that the composition of the examined copolymers is in good accordance with experimental and theoretical data regardless of the analytical method used.

Figure 4 presents high resolution C1s, O1s, and N1s core-level scans and their deconvolutions into various contributions. Details on fitting parameters are summarized in Appendix A

For all three copolymers, their C1s core-level experimental data can be decomposed into minimum six components (Figure 4a). The first component is the C=C aromatic characterized for the pendant groups with binding energies around 284.5–284.6 eV. Meanwhile, aliphatic C-C bonds are characterized by their binding energy at 285.0 eV, which has been taken as reference for calibrating the whole fitting process. The quaternary carbon of the methacrylate copolymers has higher binding energy (285.8 eV) than that of aliphatic carbon but lower than that of the C-O and/or C-N bond, whose peak maximum is around 286.3 eV to 286.8 eV. More importantly, the presence of pentafluorophenyl moieties in the copolymer caused an intense signal of the C-F bond at around 287.6 eV to 287.9 eV. The carbon of the C=O group had the highest binding among all components, at around 288.8 eV to 289.2 eV. 

Percentages of components found in the C1s core-level scan vary in function and composition. Overall, it was found that as the amount of NPMA increased, not only did the atomic percentages of carbon increase (Appendix A) but the contribution weights of carbon-to-carbon (C=C and C-C) and C-O/C-N were also greater, according to the fitting results of C1s core-level scans (Appendix A). In contrast, the percentage of C-F bonds decreased. 

As presented in Figure 4b, O1s core levels were deconvoluted into three major components including the shared C=O, C-O of carbonyl groups, and N-O characterized for NPMA. With a similarly explanation as in the case of C1s, the amount of N-O contributor increases in copolymers with higher proportions of NPMA. It is to be noted that here the full-weight half-maximum (FWHM) of C=O and C-O is quite large due to the slight difference between the carbonyl of PFPMA and that of NPMA, as observed by FTIR (i.e., absorbance bands of C=O and C-O in PPFPMA and PNPMA do not show up at the same positions). However, such derivations were neglected to simplify the fitting procedure for the XPS data.

Figure 4c,d presents N1s and F1s core-level scans of three copolymers and their deconvolutions. Each core level of all the copolymers can be deconvoluted into only one species characterized by nitro groups (around 405.8 eV) in the case of N1s and C-F bond (around 688.2 eV) in the case of F1s. From the XPS results, it is evident that the chemical structure of each copolymer corresponded to that expected from other analyses. 

### 2.4. Thermal Properties of Copolymers

The thermal properties of copolymers, especially random or statistical copolymers, are one type of indicator representing the synergic contributions of their components. In addition to its interest for polymer post-modification, PPFPMA has been copolymerized with other comonomers, specifically to improve the thermal properties of PMMA [30,31].

Figure 5a presents DSC thermograms of PPFPMA, PNPMA, and three copolymers with variations in composition. It is immediately seen that regardless of composition, each copolymer possesses only one Tg in the studied temperature range, indicating that the microstructure of copolymers is relatively even along the backbone.

In the literature, values of Tg for PNPMA obtained by free radical polymerization were reported to be 73 °C and of 195 °C, without indication of the corresponding molecular weight [32,33]. PNPMA synthesized for this work by Cu(0)-mediated RDRP, with Mn _SEC_ of around 16,200 g.mol^−1^, showed a Tg of 151.8 °C. Meanwhile, the value of Tg of PPFPMA obtained herein (i.e., 117.6 °C) was close to that reported in the literature (90 °C–125 °C) [31,34]. The deviation in obtained Tg from that reported in the literature probably resulted from the difference in molecular weight of the polymers according to Fox–Flory equation [35]. 

Figure 5b shows the relation between Tg and weight fraction of NPMA moieties. If the two monomer moieties have no interaction with each other, based on the Fox–Flory equation and experimental Tg of PPFPMA and PNPMA, as well as the weight ratio of PFPMA obtained by other analyses as discussed previously, copolymers were expected to have Tg of 126.0 °C, 132.3 °C, and 138.7 °C for copo1N2F, copo1N1F, and copo2N1F, respectively. Even though there was a linear relation between the Tg of copolymers in function of NPMA weight fraction, as shown in Figure 5b, these values are greater than expected Tg calculated from the Fox–Flory equation. Indeed, the two homopolymers were found to be of quite the same range in Mn_SEC_ (~20,000 g.mol^−1^), values half of the Mn_SEC_ of the three copolymers (~34,000–42,000 g.mol^−1^). As mentioned, such difference in Mn might be one of the reasons for the deviation in the copolymers’ Tg calculated by the Fox–Flory equation. Nonetheless, the positive derivation of Tg of copolymers in comparison to homopolymers may also rise from the interaction between two monomer moieties, as the pentafluorophenyl and nitrophenyl groups have been reported to show edge-to-face interaction [36]. Hence, the presence together of pentafluorophenyl and nitrophenyl moieties in the copolymers probably leads to new inter- and intra-molecular interactions which are absent in the case of homopolymers, resulting in positive changes in Tg of copolymers. 

Thermal stability of homopolymers and copolymers was examined by TGA under helium, as presented in Figure 6. Firstly, it was seen that the two homopolymers exhibited different thermal stability. In the case of PNPMA, the polymer possessed two major degradation temperatures. i.e., T_d, max1_ = 336 °C (~60% loss in mass) and T_d, max2_ = 431 °C (~80% loss in mass). On the other hand, PPFPMA had its first degradation zone at T_d, max1_ = 257 °C (~20% loss in mass) and the second degradation around T_d, max2_ = 368 °C (complete degradation—100% loss in mass). It is noteworthy that at 800 °C, while PPFPMA was completely decomposed, PNPMA retained 17% of its initial mass.

Due to the presence of PFPMA and NPMA monomer units, degradation temperatures of copolymers varied according to composition. It was found that the higher the content of PFPMA, the higher the degradation temperature. Overall, copo2N1F had T_d, max1_ of 343 °C, copo1N1F had T_d, max1_ of 348 °C, and copo1N2F had the highest T_d, max1_ of 350 °C. In contrast, as the content of PFPMA increased, the mass retained at 800 °C decreased. The change in thermal stability for copolymers with higher compositions of PFPMA was possibly due to the increased C-F bonding, which has higher bond dissociation energy compared with C-H, C-N, or C-O bonds. These observations can be explained by the different thermal characteristics of two homopolymers, as discussed above.

### 2.5. Post-Modification of Copolymers

To validate experimentally the reactivity of the two active ester moieties, chemical polymer post-modification of copolymers with *n*-butyl amine and citronellol was performed, as presented in Figure 2. 

Reaction of copolymers with *n*-butyl amine was investigated at room temperature in large excess (i.e., 9 eq.) of the primary amine in THF. After 24 h of reaction, a total quantitative conversion was achieved for PFPMA units by ^19^F-NMR analysis, while ^1^H-NMR analysis indicated around 55% to 65% conversion of NPMA units for all three copolymers regardless of composition. These results gave the first information on higher activity of PFPMA moieties compared to NPMA moieties. 

Furthermore, the transesterification of copo1N1F (i.e., 0.5 molar fraction of each comonomer unit) with a primary alcohol, namely citronellol (3 eq.), was carried out in either DMF or DMSO at 80 °C with 1.1 equivalent of DBU as base catalyst. ^1^H-NMR spectra of reaction carried out in DMSO and DMF after 24 h are given in Appendix A. Both reactions gave >95% (in DMSO, determined by ^19^F-NMR) to 100% (in DMF, determined by ^19^F-NMR) conversion for PFPMA units, while the substitution of NPMA units only reached 21% in DMSO and 32% in DMF as determined by ^1^H-NMR of the corresponding reaction mixture. 

The direct comparison in reactivity of PFPMA and NPMA in one single polymer chain confirmed that pentafluorophenyl moieties are much more reactive than nitrophenyl moieties towards primary amine and primary alkyl alcohol. With further in-depth investigation on reaction conditions, it is believed that this system can be used as a template for sequential post-modification. 

## 3. Materials and Methods

### 3.1. Materials

Pentafluorophenol 98% pure, tris(2-pyridylmethyl) amine (TPMA) 98%, 4,4′-dinonyl-2,2′-bipyridine (dNbpy) > 98%, methyl α-bromophenylacetate MBPA 98% (TCI, Paris, France), methacryloyl chloride 97% stabilized, *p*-nitrophenol 98%, ethyl α-bromoisobutyrate (eBiB), copper (II) bromide, tetrahydrofuran (THF) pure (Sigma Aldrich, MO, USA), 1,8-diazabicycloundec-7-ene (DBU) 98% pure) and sulfolane 99% pure (Alfa Aesar, Haverhill, MA, USA) were used as received. Copper wire 99%, d = 1.0 mm, density ≈ 7.02 g.m^−2^ (Alfa Aesar, Haverhill, MA, USA) was washed with a mixture of methanol:HCl 1M (1:1 *v*/*v*), rinsed with methanol and acetone, then dried completely before use.

### 3.2. Characterization

#### 3.2.1. NMR

^1^H, ^19^F, and DOSY NMR measurements were performed using either Bruker Advanced 250 MHz or 360 MHz (Bruker, Billerica, MA, USA) in 5-mm NMR tubes. Spectra visualization and integration were calibrated with internal signals of solvents. NMR spectra of polymers were acquired by solubilizing 15–20 mg of polymers in THF-d_8_. 

#### 3.2.2. SEC

PNPMA and copolymers were analyzed in DMF (+LiBr, 1 g.L^−1^) at Laboratory de Chimie des Polymères (LCP), Institut Parisien de Chimie Moléculaire, Sorbonne University. The analyses were performed at a flow rate of 0.8 mL.min^−1^ and toluene was used as a flow rate marker. All polymers were injected (100 μL) at a concentration of 5 g.L^−1^ after filtration through a 0.22 μm membrane. The steric exclusion was carried out on two PSS GRAM 1000 Å columns (8 × 300 mm; separation limits: 1 to 1000 kg.mol^−1^) and one PSS GRAM 30 Å (8 × 300 mm; separation limits: 0.1 to 10 kg.mol^−1^) with a Tetra detector array (TDA 305) from Malvern Panalytical (Malvern, UK), including a light-scattering detector with a right (90°) and a low (7°) angles (RALS/LALS), a laser at 670 nm, a 4-capillary differential viscometer, a differential refractive index detector (RI) and a diode array UV detector. Columns and detectors were maintained at 60 °C.

The software OmniSEC 5.12 from Malvern Panalytical (Malvern, UK) was used for data acquisition and data analysis. Molar masses (Mn, the number-average molar mass, Mw, the weight-average molar mass) and polydispersity indexes (*Ð* = Mw/Mn) were calculated with a calibration curve based on narrow poly(methyl methacrylate) (PMMA) standards (from Polymer Standard Services), using only the RI detector.

#### 3.2.3. X-ray Photoelectron Spectroscopy

A ThermoFisher K-alpha spectrometer (Thermo Fisher Scientific Inc., United States of America), equipped with a monochromatic X-ray source (Al K-alpha, 1486.6 eV), spot size 400 mm was employed; the hemispherical analyzer was operated in CAE (constant analyzer energy) mode, with passing energy of 200 eV and a step of 1 eV for the acquisition of survey spectra, and 50 eV and 0.2 eV for high-resolution spectra. A “dual beam” flood gun was used to neutralize the charge build up. The obtained spectra were analyzed with CasaXPS^®^ version 2.3.19 (Casa Software Ltd, Ħal Qormi, Malta). A Shirley-type background subtraction was used and the peak areas were normalized using the Scofield sensitivity factors in the calculation of elemental compositions. Fitting was carried out by calibrating binding energy of the C=C peak to 284.8 eV or binding energy of the C-C peak to 285.0 eV. All line shapes were considered a 30/70 or 40/60 mix of Gaussian and Lorentzian distributions.

#### 3.2.4. Elemental Analysis

Elemental analysis was performed by Perkin Elmer 2400 series elemental analyzer (Perkin Elmer, Waltham, MA, USA) at Service de Microanalyses, Institut de Chimie des Substances Naturelles (ICSN), CNRS, Gif-sur-Yvette.

#### 3.2.5. Attenuated Total Reflectance (ATR) FT-IR 

ATR FT-IR was carried out on fine powder of copolymer samples. The copolymer powder was deposited on the diamond crystals of the apparatus, then the absorbance was measured by Bruker IFS 66 equipment (Pike technologies, Fitchburg, WI, USA) with an ATR module using diamond crystals. In total, 200 scans of resolution 4 cm^−1^ were recorded between 600 cm^−1^ to 4000 cm^−1^. Spectra visualization and treatment were carried out using OPUS software (Bruker, Billerica, MA, USA); spectra integration was completed using the built-in integration function of Origin^®^ v8.0724 (OriginLab Corporation, Northamton, MA, USA).

#### 3.2.6. Thermal Analysis

Thermogravimetric analysis (TGA) measurements of copolymers were taken with a STA 449 F3 Jupiter^®^ apparatus (Netzsch, Selb, Germany). Around 5–10 mg of samples were placed in an aluminum pan, heated from 30 °C to 800 °C (heating rate 10 °C.min^−1^) in flowing helium (40 mL.min^−1^). 

Differential scanning calorimetry (DSC) measurements were conducted on a DSC 200 F3 Maia device (Netzsch, Selb, Germany). Around 5–10 mg of samples were encapsulated in sealed aluminum pans and heated from 10 °C to 200 °C at scanning rates (heating and cooling) of 10 °C min^−1^, in nitrogen atmosphere (50 mL.min^−1^). The device was previously calibrated with indium, under nitrogen atmosphere. Three of each kind of sample were measured; the glass transition temperature (Tg) was recorded in the second heating. 

### 3.3. Method

#### 3.3.1. Synthesis of PFPMA

The synthesis of PFPMA was carried out following reported protocol [4] with a few modifications as described in our previous work [19,23]. Pentafluorophenol (24 g, 0.13 mol) was dissolved in 300 mL anhydrous dichloromethane (DCM) under a moderate stream of argon. The reaction mixture was cooled down to 0 °C with an ice bath, then triethylamine (23 mL, 0.14 mol) and methacryloyl chloride (11.5 mL, 0.12 mol) were introduced dropwise subsequently via a tight argon-washed syringe. After 3 h, the reaction was left at ambient temperature overnight under argon. The reaction mixture was filtered to remove insoluble solids, then washed with saturated aqueous NaHCO_3_ solution several times to remove excess pentafluorophenol. Organic phase was dried over anhydrous MgSO_4_ before distilling under vacuum at 85 °C to collect PFPMA as transparent liquid. Yield: 80%. ^1^H NMR (250 MHz, CDCl_3_, δ/ppm): 2.06 (s, 3H, CH_3_), 5.89 (s, 1H, CH_2_=C), 6.43 (s, 1H, CH_2_=C). ^19^F NMR (250 MHz, CDCl_3_, δ/ppm): −162.45 (t, 2F, meta positions), −158.14 (t, 1F, para position), −152.75 (d, 2F, ortho positions).

#### 3.3.2. Synthesis of NPMA

The synthesis of NPMA was carried out following reported protocol [8]. *p*-Nitrophenol (5 g, 0.036 mol) was dissolved in 120 mL of anhydrous dichloromethane under a flow of argon. Triethylamine (5.5 mL, 0.040 mol) was added dropwise into the reaction mixture. The reaction flask was then placed on an ice bath to cool the mixture to 0 °C before methacryloyl chloride (3.6 mL, 0.032 mol) was introduced dropwise via a syringe. The reaction mixture was left stirring at 0 °C for 3 h then at room temperature overnight. The reaction mixture was filtered through filter paper to remove white salt. The collected organic phase was then transferred into a 500 mL separatory funnel, followed by the addition of 25 mL HCl 1M then washed twice with NaOH 1M, rinsed with 15 mL water and dried by anhydrous MgSO_4_. The organic phase was filtered in a Buchner funnel to collect liquid phase which was then concentrated under vacuum. The obtained product was dissolved in a minimum volume of dichloromethane and *p*-nitrophenyl methacrylate crystals were then collected by precipitation in petroleum ether. Yield: 40%. ^1^H NMR (250 MHz, CDCl_3_, δ ppm): 2.08 (s, 3H, -CH_3_), 5.85 (s, 1H, CH_2_=C), 6.40 (s, 1H, CH_2_=C), 7.32 (d, 2H, *J* = 9.2 Hz, ortho-arom. protons), 8.29 (d, 2H, *J* = 9.2 Hz, meta-arom. protons). 

#### 3.3.3. Cu(0)-mediated RDRP of PFPMA and NPMA

Prior to use, 1 cm copper wire was washed with a mixture of HCl (1M) and MeOH (1:1 *v*/*v*), rinsed with acetone, and dried under vacuum. In a round bottom flask, NPMA at given weight was dissolved in 1320 μL THF, then a mixture of TPMA (4.3 mg, 0.015 mmol) and CuBr_2_ (0.8 mg, 3.6 × 10^−3^ mmol) in 660 μL of sulfolane was introduced, followed by the introduction of PFPMA. The mixture was degassed for 15 min under argon. Copper and eBiB (5.0 μL, 3.4 × 10^−5^ mmol) were introduced respectively. The flask was sealed with a rubber septum and parafilm before immersion in a preheated oil bath at 60 °C. The reaction was stopped at desired time. Reaction mixture was passed through a basic aluminum column with DCM as solvent. The collected solution was then concentrated and precipitated in excess MeOH. Further purification was carried out by redissolution of copolymers in THF, reprecipitation in excess MeOH, and drying under high vacuum at room temperature.

#### 3.3.4. Post-Modification of Copolymers

In a dry round bottom flask, under a moderate stream of argon, 250 mg of copolymer was dissolved in 1 mL of either anhydrous THF, DMSO, or DMF. A desired equivalent of dry citronellol (3 eq.) or *n*-butyl amine (9 eq.) was then rapidly introduced. In the case of alcohol, DBU (1.1 eq.) was added as base catalyst. The flask was then sealed under argon and immersed in an oil bath at 60 °C. Conversion for PFPMA moieties was calculated from ^19^F-NMR spectra of the reaction mixture. Conversion of NPMA moieties was calculated from ^1^H-NMR spectra of the reaction mixture.

## 4. Conclusions

In conclusion, copolymerization of NPMA and PFPMA via Cu(0)-mediated RDRP was successfully performed with high to very high comonomer conversion and acceptable dispersity for controlled polymerization. Chemical composition and characteristics of each copolymer were analyzed using several techniques including NMR, FTIR, SEC, elemental analysis, XPS, DSC, and TGA. All of the obtained results provided excellent agreement on chemical composition. It was found that the molar ratio between comonomers in the final product was very close to that in the feed, confirming the comparable reactivity during polymerization. In addition, thermal analysis of three obtained copolymers revealed high glass-transition temperature (Tg > 140 °C) and relatively good thermal stability. The success of this work implies the versatility of Cu(0)-mediated RDRP in copolymerization of similarly structured monomers. Furthermore, as the post-modification of copolymers confirms higher reactivity of PFPMA moieties compared to NPMA moieties, the use of copolymers derived from both PFPMA and NPMA can eventually open up new opportunities to synthesize better controlled functional copolymers based on sequential polymer post-modification.

## Data Availability

Data are contained within the article and Appendix A.

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
