# Peer review of "Copolymers Derived from Two Active Esters: Synthesis, Characterization, Thermal Properties, and Reactivity in Post-Modification"

_molecules, 2022, doi:10.3390/molecules27206827_

Round 1

Reviewer 2 Report

Statistical copolymers of pentafluorophenyl methacrylate and p-nitrophenyl methacrylate were prepared via Cu(0)-mediated reversible deactivation radical copolymerization. It is an interesting work, which is rather well executed in terms of synthesis and characterization of the desired products. However, this work is not completed and needs further data to be accumulated in order to have a full description of the copolymerization system. What is normally expected from these copolymerization studies is the determination of the copolymerization reactivity ratios, the dyad sequences of the monomer units, the mean sequence length etc. For this reason, it is important to achieve low copolymerization conversions in order to employ the copolymerization equation and then apply linear and non-linear methodologies for the determination of the reactivity ratios. Having these results, it will be easier to understand the DSC and TGA data employing suitable models for the prediction of the thermal properties. It looks strange to have Tg values of the copolymers much higher than those of the respective homopolymers. To ensure that these differences are due to differences in the molecular weight of the samples, the authors should synthesize samples of similar molecular weights. The TGA analysis is also incomplete, since the data are not connected with the mechanism of the thermal decomposition. The first derivative of the TGA plots (Differential ThermoGravimetry, DTG) is better to be employed for comparison of the various samples. Finally, careful editing in the text is needed.   

Round 2

Reviewer 1 Report

The corrections developed on the manuscript allow its publication to be accepted.

Referee

Reviewer 2 Report

I can understand the arguments of the authors regarding the suggestion to perform extra work for this manuscript and I recognize their effort to improve the quality of their work. Therefore, I suggest the publication of this manuscript in Molecules.